

# Food taboos during pregnancy: meta-analysis on cross cultural differences suggests specific, diet-related pressures on childbirth among agriculturalists

Ornella Maggiulli[1], Fabrizio Rufo[1], Sarah E. Johns[2] and Jonathan C.K. Wells[3]

[1] University of Roma "La Sapienza", Rome, Italy
[2] University of Kent, Canterbury, United Kingdom
[3] University College London Great Ormond Street Institute of Child Health, London, United Kingdom

Corresponding author
Ornella Maggiulli,
ornella.maggiulli@uniroma1.it

## ABSTRACT

Pregnancy is the most delicate stage of human life history as well as a common target of food taboos across cultures. Despite puzzling evidence that many pregnant women across the world reduce their intake of nutritious foods to accomplish cultural norms, no study has provided statistical analysis of cross-cultural variation in food taboos during pregnancy. Moreover, antenatal practices among forager and agriculturalists have never been compared, despite subsistence mode being known to affect staple foods and lifestyle directly. This gap hinders to us from understanding the overall threats attributed to pregnancy, and their perceived nutritional causes around the world. The present study constitutes the first cross-cultural meta-analysis on food taboos during pregnancy. We examined thirty-two articles on dietary antenatal restrictions among agricultural and non-agricultural societies, in order to: (i) identify cross-culturally targeted animal, plant and miscellaneous foods; (ii) define major clusters of taboo focus; (iii) test the hypothesis that food types and clusters of focus distribute differently between agricultural and non-agricultural taboos; and (iv) test the hypothesis that food types distribute differently across the clusters of taboo focus. All data were analysed in SPSS and RStudio using chi-squared tests and Fisher's exact tests. We detected a gradient in taboo focus that ranged from no direct physiological interest to the fear of varied physiological complications to a very specific concern over increased birth weight and difficult delivery. Non-agricultural taboos were more likely to target non-domesticated animal foods and to be justified by concerns not directly linked to the physiological sphere, whereas agricultural taboos tended to targed more cultivated and processed products and showed a stronger association with concerns over increased birth weight. Despite some methodological discrepancies in the existing literature on food taboos during pregnancy, our results illustrate that such cultural traits are useful for detecting perception of biological pressures on reproduction across cultures. Indeed, the widespread concern over birth weight and carbohydrate rich foods overlaps with clinical evidence that obstructed labor is a major threat to maternal life in Africa, Asia and Eurasia. Furthermore, asymmetry in the frequency of such concern across subsistence modes aligns with the evolutionary perspective that agriculture may have exacerbated delivery complications. This study highlights the need for the improved understanding of dietary behaviors during pregnancy across the world, addressing the

role of obstructed labor as a key point of convergence between clinical, evolutionary and cultural issues in human behavior.

## INTRODUCTION

Taboos have been described as means of preserving local consensus about the organization of the world to hedge around dangers arising from ambiguous or anomalous sources (*Douglas, 2003*), and they often address life-course milestones or vulnerable life stages. Recognized as a period of fragility and higher mortality risk, pregnancy is a common target of behavioural as well as nutritional norms that are reinforced by the larger community to ensure a successful delivery and what is locally considered a normal process of development (*Lee et al., 2009*; *Nag, 1994*; *Spielmann, 1989*; *Ebrahim, 1980*; *Kuzma et al., 2013*; *Pagezy, 2006*; *Daviau, 2003*; *Aunger, 1994*; *Carles, 2014*; *Marcel, Justine & Florentine, 2015*; *Du Jeune, 2014*). Childbirth is indeed likely to have been a major source of mortality until the advent of modern medicine (*Trevathan, 2010*, p. 98) and remains so in many non-industrialized settings (*Hogan et al., 2010*).

From demographic and evolutionary perspectives, pregnancy acts as ''a funnel through which all individuals must pass,'' as constrained maternal nutritional, energetic, physical, and immunological requirements to support the foetus lead to a sensitive life-stage that is particularly sensitive to selective pressures (*Brown, Ruvolo & Sabeti, 2013*). Evidence that pregnant women in various parts of the world are often forced to abstain from especially nutritious foods during such a delicate life-stage (*Meyer-Rochow, 2009*; *Bentley et al., 1999*), indicates that antenatal dietary restrictions remain a relatively unexplored anthropological 'puzzle' that deserves major academic attention (*Rush, 2000*). Arguably the prescription most recognised in the biomedical literature is the advice to 'eat down' during pregnancy or at least to not exceed the normal dietary intake, which has frequently been described in Asian countries, and more rarely the advice to 'eat up' (*Brems & Berg, 1988*; *Asim et al., 2021*; *Harding et al., 2017*; *Christian et al., 2006*). However, although it is often considered that eating down is recommended to achieve an easy delivery, in some populations the practice has been primarily attributed to women feeling unwell in late pregnancy (*Asim et al., 2021*; *Harding et al., 2017*; *Karim et al., 2002*; *Christian et al., 2006*; *Brems & Berg, 1988*; *Choudhry, 1997*; *Nichter & Nichter, 1983*; *Nag, 1994*; *Holmes et al., 2007*; *Hutter, 1996*; *Rush, 2000*; *Shannon et al., 2008*; *Yeasmin & Regmi, 2013*). Our focus here deliberately goes beyond dietary energy content *per se*, and focuses on individual types of foods that may have specific metabolic effects relevant to childbirth complications or other outcomes.

Before introducing any possible adaptive perspective on cultural norms that regulate dietary behaviour during pregnancy, it is important to note that symbolic and functionalist

approaches alone cannot provide a compelling account of any kind of food taboo (*Navarrete & Fessler, 2003*; *Simoons, 1994*). Rather, it is necessary to reconcile, conceptually, the symbolic, social and ecological value of taboos (*Golden & Comaroff, 2015*; *Meyer-Rochow, 2009*; *Osei, 2006*), even considering that the health value of taboos may be indirect or not obvious. The strong social functions and religious motivations that maintain food taboos in a given culture do not exclude the fact that they also often target recognized agents of zoonotic disease risk or allergic responses (*Pagezy, 2006*; *Golden & Comaroff, 2015*; *Meyer-Rochow, 2009*). With regard to pregnancy, reasons that bring women to engage in taboos range from meeting family and social expectations to the point of "disagreeing but complying" (*Chang, Kenney & Chao, 2010*), to the internalized belief that taboos are beneficial to the mother and the baby (*Otoo, Habib & Ankomah, 2015*). Perceptions of wellbeing vary dramatically across cultures (*Napier, 2015*), and the intervention of non-material forces are commonly seen as both causal agents of illness/death and the threats that taboos should protect from (*Carles, 2014*; *Bolton, 1972*; *Townsend, 1971*). Therefore, taboos during pregnancy, as well as taboos focused on other life-stages, should be approached under the expectation that health and non-health intents related to the periods before, during and after childbirth may not be immediately distinguishable according to the western concept of "health". However, we propose that, within an overarching framework of social and symbolic values, food taboos during pregnancy are particularly likely to reflect specific biological pressures, at least at a cross-cultural scale. Compared to dietary restrictions applied to other life events, greater attention should be paid to the fitness implications and maternal risks associated with pregnancy, and the consequent nature of childbirth as major agent of natural selection (*Trevathan, 2010*, p. 98).

Food aversions and fasting may be perceived to serve different purposes in different times and populations, making it important to increase our understanding of the specific reasons provided in any specific setting. *Marcel, Justine & Florentine (2015)* and *Arzoaquoi et al. (2015)* observed that antenatal dietary restrictions are often driven by vital and preventive intents that are distinct from 'aesthetic' reasons to ensure the attractiveness of the unborn child according to local standards, and social pressures to respect tradition. This is consistent with fragmented evidence that dietary taboos for pregnant women may variously aim to avert food poisoning (*Henrich & Henrich, 2010*), abortion (*Placek, Madhivanan & Hagen, 2017*), big babies (*Rush, 2000*), or congenital malformations (*Holmberg, 1950*). Early evolutionary frameworks on maternal diet and food aversions were advanced under the perspective that pregnancy sickness evolved to prevent women from ingesting meat-carried pathogens and teratogens contained in plants during the early stage of pregnancy, *i.e.*, when the organogenesis takes place (*Flaxman & Sherman, 2000*; *Profet, 1992*). Despite meat being a common target of both food aversions and taboos, however, this is not sufficient to affirm the former drive, or fully explain, the latter (*Fessler, 2002*). *Placek et al. (2021)* have found that women do not fast according to the maternal-fetal protection hypothesis suggested above during Ramadan, but perhaps to gain moral capital. However, Ramadan is a universal ban that eventually impacts pregnant women, therefore it seems less informative when exploring the evolution of pregnancy dietary behaviour, than taboos evolved purposely to control the pregnancy output. Indeed, there is a lack of comprehensive evaluation of

avoided foods, worldwide, during pregnancy and the reasons for their avoidance. This hinders the opportunity to apply a cross-cultural approach to women's nutrition during pregnancy as a tool to identify clusters of feared complications, their perceived relation to diet, and therefore, to achieve a robust evolutionary perspective on antenatal taboos.

Early attempts to compare cross-cultural food taboos during pregnancy helped to pinpoint some commonly avoided food types and the reasons for the taboo (*Iradukunda, 2020*; *Carles, 2014*), but these studies did not provide strong statistical evidence of the cross-cultural variance, nor of associations between avoided food categories and the underlying reasons. Moreover, dietary restrictions during pregnancy among hunter-gatherers were not included in any available cross-cultural comparison.

We argue that to investigate cross-cultural, diet-attributed undesired pregnancy complications, information on subsistence mode is particularly relevant. Subsistence patterns strongly affect the balance of macronutrients in the diet, and the different ratio of plant to animal foods between agricultural and foragers' diets represents the most marked nutritional discrepancy across human evolution. In recent millennia, agriculture and industrialization have introduced cultivated starches, cereals, and processing methods that increased the glycaemic load of human diet. In particular, the estimated carbohydrate intake of current agricultural societies ranges from 65 to 91% of total energy intake, whereas that of current foragers does not exceed 22–40% (*Ulijaszek, 1995*; *Ulijaszek, Mann & Elton, 2012*; *Cordain et al., 2000*; *Cordain et al., 2005*; *Brown, Ruvolo & Sabeti, 2013*). Such a marked change in diet and lifestyle has altered metabolic patterns and exposure to disease across populations (*Ulijaszek, Mann & Elton, 2012*). *Voeks & Sercombe (2000)* observed differences between the ethnomedical systems of cultivators and foragers of Brunei, arguing that such systems can be a direct function of subsistence modes and their intrinsic capacity to differentiate a population's exposure to environmental threats. Therefore, under the assumption that food taboos during pregnancy are likely to reflect biological pressures, we hypothesise that different subsistence patterns, by producing different staples in different quantities, determine different food taboos and for diverse reasons that remain under-explored.

In the present study, we analysed current literature and ethnographic accounts relating to food taboos during pregnancy, in order to: (i) identify cross-culturally targeted animal, plant and miscellaneous foods; (ii) define major clusters of taboo focus, by distinguishing concerns over physiological threats from those considered to mitigate non-physiological aspects; (iii) test the hypothesis that food types and clusters of focus distribute differently between agricultural and non-agricultural pregnancy-related dietary taboos; and (iv) test the hypothesis that food types distribute differently across clusters of taboo focus. We detected a gradient in taboo focus that ranged from no direct physiological interest to a very specific concern over increased birth weight and difficult deliveries, as well as a different distribution of food taboos during pregnancy across subsistence modes. We propose a post-hoc interpretation of this latter result, namely that carbohydrate-rich diets may generate major concern over high birthweights and difficult delivery, which may help explain the clustering of taboos around this issue in agricultural societies.

## METHODS

We searched literature on food taboos and dietary practices during pregnancy on Scopus, Google Scholar, PubMed and eHRAF World Cultures in English, Spanish and French. Search queries were: "Food taboos/avoidances during pregnancy/gestation", "Dietary restrictions/behaviour during pregnancy/gestation". The eligibility criteria were to select articles that contained local/common names or food categories avoided during pregnancy and reasons for the avoidance in cultural contexts not biased by Western medical guidelines. Articles that mentioned food taboos but did not give reasons for the taboos were not selected while, within the selected articles, food taboos that were mentioned without reasons for the ban were excluded from statistical analysis (Fig. 1). Thirty-two articles met the eligibility criteria and included evaluations specifically of food taboos during pregnancy as well as on generic taboos with a mention of dietary restrictions during pregnancy. Eleven articles referred to non-agricultural contexts, while the remaining twenty-one addressed agricultural practices. Among original articles, ten provided a list of food taboos during pregnancy that was derived from quantitative studies, while twenty-one qualitative studies were based on interviews on samples that ranged in size from 11 (*Pearn & Sweet, 1977*) to 1,200 (*Ferro-Luzzi, 1973*) respondents.

Our methodologic approach was aimed at maximizing the value of the data currently available on antenatal practices, by investigating which food-stuffs are thought to be dangerous during the human's most delicate life history stage and why, and also among which food production pattern. In line with the basic guidelines of cross-cultural research (*Ember & Ember, 2011*) we selected only descriptive data on taboo food names and reported reasons among a given human group, retaining older studies from non-agricultural contexts if they were the only sources available for particular regions on the specific subject of antenatal dietary practices (*e.g.*, *Spencer & Gillen, 1898*; *Brown, 1922*; *Roheim, 1933*). These eligibility criteria were applied to ensure that statistical analyses were conducted only on the most accurate descriptive data on both tabooed foods and underlying reasons available in the literature. Future research could investigate, and potentially augment through novel ethnographic projects, the remaining literature on maternal dietary behaviour that was excluded from this study because of its incomplete contribution in terms of detail on avoided food and reasons. We managed to obtain descriptive data on food taboos during pregnancy and reasons for at least one cultural context from almost all the major geographical regions classified in the Outline of World Cultures–OWC (*Murdock, 1983*), *i.e.*, Asia, Africa, North America, South America, Oceania and Eurasia, with the exception of Europe and Middle East (Table 1). This ensured that our comparative analyses respected the criteria of maximum geographical dispersal (*Ember & Ember, 2011*) within the limits of current available descriptive data on antenatal dietary practices. Table 1 shows the thirty-two references used to build our database of taboo food types and reasons together with the localization and subsistence pattern of the population under study. The variables of interest, analysed in SPSS (Version 27.0), were *Subsistence pattern*, *Food name*, *Food type*, *Taboo reason*, and *Taboo focus*, and we coded the data as described below.

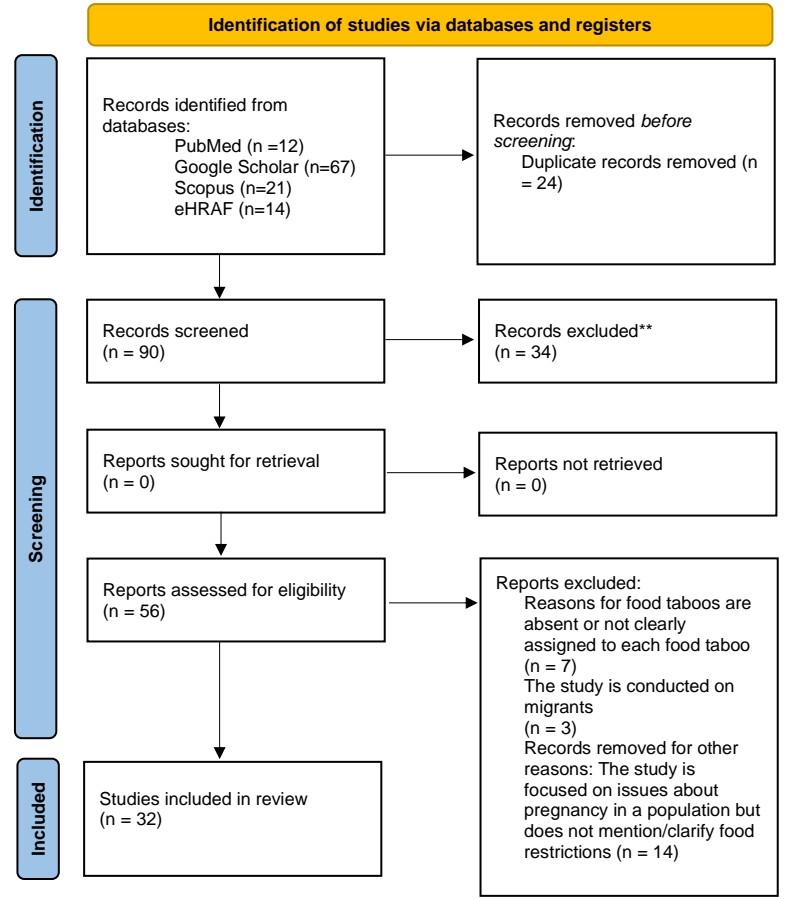

**Identification of studies via databases and registers**

**Identification**

Records identified from databases:
  PubMed (n =12)
  Google Scholar (n=67)
  Scopus (n=21)
  eHRAF (n=14)

Records removed *before screening*:
  Duplicate records removed (n = 24)

**Screening**

Records screened (n = 90)

Records excluded** (n = 34)

Reports sought for retrieval (n = 0)

Reports not retrieved (n = 0)

Reports assessed for eligibility (n = 56)

Reports excluded:
  Reasons for food taboos are absent or not clearly assigned to each food taboo (n = 7)
  The study is conducted on migrants (n = 3)
  Records removed for other reasons: The study is focused on issues about pregnancy in a population but does not mention/clarify food restrictions (n = 14)

**Included**

Studies included in review (n = 32)

**Figure 1** **PRISMA flow diagram.**

## Subsistence pattern

*Subsistence pattern* refers to the main mode of food production of the communities whose food taboos during pregnancy were analysed in the original articles. Agriculture impacts macronutrient balance substantially, compared to other strategies, in terms of the supply of plant foods and carbohydrate intake (*Cordain et al., 2005*). Because we aimed to investigate diet-related cross-cultural differences in food taboos during pregnancy, particularly focusing on the increase in glycaemic load as the main dietary shift associated with agriculture, we relied on agricultural systems as the main source of food supply as a discriminating factor to create two categories of subsistence patterns: *Agriculture*, including both subsistence and intensive agriculture, and *Non-agriculture*, encompassing strategies where agriculture was absent or marginally practiced with regard to foraging or fishing. This allowed us to make an initial comparison of taboos during pregnancy between societies potentially exposed to higher versus lower glycaemic load, since the increasing carbohydrate/protein ratio driven by agriculture represents a key dietary shift in human evolution. However, we recognize that the specific subsistence strategies aggregated here in a binary categorization may be linked with equally specific diets and potential

**Table 1** References on food taboos during pregnancy showing the location and subsistence pattern of the human groups under study.

| Non-agricultural societies | Agricultural societies |
|---|---|
| *Holmberg Allen, 1950* (Bolivia) | *Demissie & Kogi-Makau, 1998* (Ethiopia) |
| *Aunger, 1994* (Zaire) | *Arzoaquoi et al., 2015* (Ghana) |
| *Pagezy, 2006* (Congo) | *Dove, 2010* (Ghana) |
| *Brown, 1922* (Andamane) | *Olarinoye et al., 2014* (Nigeria) |
| *Meggitt, 1965* (Australia) | *Odebiyi, 1989* (Nigeria) |
| *Pearn & Sweet, 1977* (Australia) | *Ekwochi et al., 2016* (Nigeria) |
| *Spencer & Gillen, 1898* (Central Australia) | *Donné Kouadio, 2017* (Ivory Coast) |
| *Roheim, 1933* (Central Australia) | *Marchant et al., 2002* (Tanzania) |
| *Townsend, 1971* (Papua New Guinea) | *Mustafina et al., 2019* (Kazakhstan) |
| *Bolton, 1972* (Malaysia) | *McNamara & Wood, 2019* (Tajikistan) |
| *Henrich & Henrich, 2010* (Fiji) | *Shannon et al., 2008* (Bangladesh) |
| | *Ferro-Luzzi, 1973* (India) |
| | *Daviau, 2003* (Lao) |
| | *Holmes et al., 2007* (Lao) |
| | *Kuzma et al., 2013* (Papua New Guinea) |
| | *Lepowsky, 1985* (Papua New Guinea) |
| | *Liamputtong et al., 2005* (Thailand) |
| | *Gao et al., 2013* (China) |
| | *Hartini, 2004* (Indonesia) |
| | *Oishi et al., 2000* (Korea, Japan) |
| | *Pritham & Sammons, 1993* (Korea) |

hazards to pregnant women. Future research should investigate these specific links either by building a larger ethnographic cross-cultural database, or through in-depth studies on economic factors and foetal growth patterns within a given geographical area. *Subsistence pattern* was treated as a nominal variable, numerically coded with the following values: 1 = *Agriculture*, 2 = *Non-agriculture*. Taboos belonging to communities relying on *Agriculture* and *Non-agriculture* are hereafter referred to as agricultural taboos and non-agricultural taboos.

## Food name

*Food name* refers to foods subject to taboos for pregnant women, as described in the original articles. They include local or common names of animals, plants, and processed and miscellaneous food (Table 2). *Food name* was treated as a nominal variable with string type items.

## Food type

*Food type* refers to the food categories that best describe *Food name* items. Since subsistence and intensive agriculture increases the presence of both cultivated plants and processed foods in the diet, we defined three major categories of food types to describe food taboos: *Animal products*, including animals, animal parts, milk, eggs and honey; *Plant products*, including vegetables, tubers, cereals, pulses, seeds, fruits, and plant-based non composite

Maggiulli et al. (2022), *PeerJ*, DOI 10.7717/peerj.13633

**Table 2  Food taboos among agricultural and non-agricultural subsistence patterns, stratified by food type and taboo focus.** Frequency values of food taboos higher than 1 are indicated in parenthesis.

| | Agricultural taboos | | |
| --- | --- | --- | --- |
| | **Big baby and/or difficult delivery** | **Varied physiological complications** | **No or unspecified physiological conditions** |
| **Animal products** | Eggs (3), milk (2), fish (2), buffalo milk, bush meat, cat, cheese, farmed meat, fatty meat, fresh meat, ghee, grasscut, head of animals, horse, liver, meat, meat sauce | Fish (3), catfish (2), crab (2), camel, eel, shark, squid, shellfish, masi (fermented fish preparation), chicken, civet, goat, honey, lamb, lungs, meat, mutton, pork, rabbit, snail, snake, soft shell turtle, sow, wild animals | Snake (3), eggs (3), duck (3), snail (3) chicken (2), crab (2), dog (2), octopus (2), rabbit (2), bacon, bandicoot, beef, flying fox, fowl, goat, pigeon, shark, squid, stewed meat, tree kangaroo |
| **Plant products** | Banana (4), sugarcane (2), aloco, Bengal gram, breadnut, cassava, corn flour, enset bread, garri, kale, leaves of candlenut tree, lentils (dhal), linseed, mango, nuts, orange, osh (rice dish), pistachios, potato, shiro wot, starchy foods, sweet potatoes, taro, taro Singapore, teff injera | Eggplant (2), aibika leaves, ash pumpkin, bambara beans, banana, bottlegourd, durian, ebolo, fermented cassava/rice/sticky rice, ginger, groundnuts, horsegram, jackfruit, jiggery, lime, mango, mustard seeds, black nightshade, palm sugar, palmyra fruit, papaya, pineapple, pumpkin, rice crust, sesame, spices, sugarcane, tamarind, taro, tinai millet, tubers, vegetables | Rice (2), anise, black grapes, chillies, clove, coconut milk, custard apple, eggplant, fried maize, jambu fruit, leek, plantain, sauerkraut, wheat, yellow marita |
| **Processed and miscellaneous products** | Noodles (2), baked goods, bread, cold and sugary foods, dumplings, foods with carbohydrate, ice cubes, leftovers, protein food, shea butter, sour foods, wheat bread | Salt (2), cool foods, hot and spicy food, ice cream, leftovers, oily foods, pickled food, spicy food, sugar | Coffee, tea, hot pot |
| | Non-agricultural taboos | | |
| **Animal products** | Too much meat | Djufia snake (2), antilope Bongo, blue-coloured fish, coati, duikers, honey bear, jaguar, large fish, porcupine, potamocherus, sardines, situtunga, slippery animals, turtles' eggs, unborn dugong foetus, water chevrotain, water-snake, snake | Civet (2), large fish (2), lizards (2), all meat, animals, anteater, armadillo, bamboo rat, bat, bear, cassowary, , crocodile, deer, dugong, frog, full grown pig, full grown turtle, gibbon, guan, haru werio snake, hawks and eagles, honey, hornbill, kangaroo, komar fish, leaf monkey, longtailed macaque, loris, maeaw, octopus, otter, owl, owl monkey, pigtailed macaque, porcupine, pork, rat, squirrel, tortoise, toucan, turtle, wild pig, skin |
| **Plant products** | | Shiitake mushroom | Deformed plants, double ear of corn or manioc, yam |

dishes; *Processed and miscellaneous products*, including plant origin foods that are highly processed (*e.g.*, baked goods), composite foods (*e.g.*, dumplings), industrial products (*e.g.*, sugar, industrial drinks), foods referred to with a generic connotation (*e.g.*, fatty foods, leftovers), and foods that do not belong to any previous category (*e.g.*, coffee, ice cubes) (Table 2). Each item of the *Food name* variable was assigned to a *Food type* nominal category, numerically coded with the following values: 1 = *Animal products*, 2 = *Plant products* and 3 = *Processed and miscellaneous products*.

### Taboo reason

*Taboo reason* refers to the reason for avoidance given for each food taboo in the original articles. *Taboo reason* was treated as a nominal variable with string type items.

### Taboo focus

*Taboo focus* is the variable that reduces the information represented by all the individual reasons given for a taboo into a smaller number of categories, defined only after the collection of all the data. Our initial expectation was to be able to distinguish non-physiological concerns from diverse clusters of specific physiological concerns. Indeed, we observed that the reasons were classifiable through a qualitative gradient in the physiologic value, likelihood and specificity of the risk attributed to the consumption of taboo foods. We defined *a posteriori* the following three categories of taboo focus: (1) *No or unspecified physiologic complications*, encompassing undesired events like generic "illness" and those not, or less obviously, linked to actual physiologic danger (*e.g.*, "the baby becomes fearful like an armadillo if the mother eats it", or other features despised at a locally defined aesthetical level); (2) *Varied physiologic complications*, including undesired conditions that were more specific and plausible from a physiological point on view, independently from scientifically valid relations of causality with taboo foods, and whose individual frequency was not high enough to build a separate category (*e.g.*, "miscarriage", "nausea"); and (3) *Big babies and/or difficult delivery*, which refers to a numerically consistent group of specific concerns over prolonged labour, often in association with a feared high birth-weight of the baby. Each item of the *Taboo reason* variable was assigned to a *Taboo focus* nominal category, numerically coded with the following values: 1 = *Big babies and/or difficult delivery*, 2 = *Varied physiologic complications*, and 3 = *No or unspecified physiologic complications* (Table 3).

### Statistical analyses

We performed chi-squared tests via 3 × 2 cross-tabulation in SPSS (Version 27.0) to test associations between *Food type* and *Subsistence patterns*, and between *Taboo focus* and *Subsistence patterns*, using a threshold of $p = 0.05$ to define statistical significance and Cramer's V to measure effect size. After testing whether the variables were associated, we performed $z$-tests on column proportions (*i.e.*, on the count in a single crosstab cell divided by the base of the related column) to investigate which of the three categories of *Food type* and *Taboo focus* (rows) differed significantly between the two *Subsistence pattern* categories (columns). We applied chi-squared and $z$-tests via cross-tabulation also to test

Maggiulli et al. (2022), *PeerJ*, DOI 10.7717/peerj.13633

**Table 3 Reasons for avoidance of food during pregnancy as reported in the sources, divided by categories of taboo focus.** Frequency values of reasons higher than 1 are indicated in parenthesis.

| Big baby and/or difficult delivery | Varied physiological complications | No or unspecified physiological complications |
|---|---|---|
| Big baby (25); Excessive weight-gain and a risky delivery (8); Difficult delivery (6); Excessive weight and difficult delivery (5); Excessive weight, difficult delivery and possible death of the mother (4); Upside down baby during delivery (2); Late difficult delivery, the baby doesn't want to born (2); Baby doesn't want want to born, big tummy; Baby will grow too big causing complications during labour; Baby will grow too big causing obstructed labour; Birth obstruction; Complicated delivery; Difficulties during labour; The child will be too big, and it will be difficult for him to come out; Excessive weight; Heavy bleeding at delivery; Haemorrhage during delivery due to sugar; Haemorrhage and painful delivery; Late and difficult delivery; Pain during delivery | Miscarriage (30); Bleeding (4); Epilepsy (4); Gastrointestinal problems (3); Allergy (2); Clubfooted child (3); Excessive belching (2); Nausea (2); Nausea or disgust (2); Respiratory problems (2); Skin problems (2); The baby has fits (2); Amplified morning sickness; Anal pain after delivery; Asphyxia; Baby deformity; Baby develops cough; Big placenta; Cleft-lip; Death; Deformed or paralyzed baby; Deformity; Disability; Drooling; Easier to get bacterial infection of the skin for the baby; Excessive drooling; Feces during delivery; Generalized weakness; Giddiness in the mother, skin disease, beriberi and fits in the baby; Hairless; Headaches and itching; Heartburn; Mother will have difficulties to walk; Mouth injury; Respiratory and skin problems; Shortness of breath; Skin allergy and sputum; Skin disease; Skin problems, sickness, death; Soares and very long head; Sticky placenta; Stillborn; Swollen feet; Pregnancy could last up to twelve month; The perineum does not dry out properly after birth; They produce gas in the woman; Vascular pain; Toe abnormalities; Weakness of limbs | Animal's too strong spirit causes severe illness (24); The child takes the characteristics of the animal (13); Illness for child or parents (8); Harmful due to lot of flavour, not fresh enough (7); Mother killed by the spirits (4); The plant transmits its dark colour to the baby (3); Baby cries a lot at birth (3); Harmful (2); Sore eyes (2); No skeleton (2); Unintelligent (2); Too cold (2); Too hot (2); Baby can develop a hand like the animal; Baby skin will be red like burned; Baby will have skin with scales like a snake; Baby would cry like flying fox; Formation of sticky layer of fat around the newborn; Chickenlike skin; Child behaves like a dog or mute; Child does not sleep at night; Child only has two descendants; Baby has coarse skin like sharks; Dangerous birth; Foetal abnormality; Hairless child (like an egg); Baby has fear like an armadillo; Hormonal changes; Horns and webbed feet (like ducks); Hyperactivity; Lewdness; Long delivery like chickens; Not appreciated in general; Rice will stick to baby's skin; Sickness; The baby becomes sluggish and salivate excessively like a snail; Twins (due to the duality of the doubled plants); Ungainly long legs; Unsightly large mouth; Wild child like dogs |

**Table 4  Significant differences in avoided food types during pregnancy across subsistence modes.** The values (frequencies and percentages) in bold type are those that are significantly higher than the values in italics in the same row.

|  |  | Agricultural taboos | Non-agricultural taboos |
|---|---|---|---|
| Animal products | N (%) | *81 (42.9%)* | **70 (94.6%)** |
| Plant products | N (%) | **82 (43.4%)** | *4 (5.4%)* |
| Processed/miscellaneous products | N (%) | **26 (13.8%)** | *0 (0%)* |
| Totals | N (%) | 189 (100%) | 74 (100%) |

whether *Food type* (rows) was associated with *Taboo focus* (columns) and to investigate which food type categories differed significantly between taboo focus categories. In the results section, we report column percentages (column proportions multiplied by 100) for each category of *Food type* and *Taboo focus* derived from SPSS contingency table outputs. We performed Fisher's exact test on differences in *Food type* and *Taboo focus* between *Subsistence patterns* within each geographical region for which data on both subsistence patterns were available, in order to control for geographical bias in the distribution of food taboos (*i.e.*, to exclude that a "super group" of *e.g.*, agriculturalists from one geographic area could overshadow the concerns of agriculturalists from other regions, or that agriculturalists and non-agriculturalists from the same region may have the same taboos due to cultural continuity, that are not detected in the aggregation of the data). The geographical classification remained that of the Outline of World Cultures–OWC (*Murdock, 1983*) because this classification guided the bibliographical research to ensure a geographically-varied sample in the first place. The geographical regions for which data on both subsistence patterns were available were Africa (Nigeria, Ghana, Tanzania, Ethiopia, Ivory Coast, Congo), Asia (Bangladesh, Thailand, Korea, China, Malaysia, Andaman Islands, Japan, Indonesia, India, Laos) and Oceania (Fiji, Papua New Guinea, Australia). A comparison between each nationality or small regions, instead of larger geographic areas, would have not been informative for this database, because the number of nationalities would have been too high compared to the data available for each. Future research needs to overcome this limitation by providing accurate data on subsistence and maternal dietary patterns for a wider number of cultures. Given the smaller sample of the control analysis compared to the main database (one geographical area, one test), a Fisher's Exact test  was performed in RStudio (*RStudio Team, 2021*).

## RESULTS

### Overview

We detected thirty-two eligible studies and 263 food taboos with related reasons in the current literature on antenatal dietary practices. Studies that were not eligible included: articles that mentioned food taboos during pregnancy but reasons for food taboos were absent or not clearly assigned to each food taboo (*Trigo et al, 1989*; *Pezzuti, 2004*); and studies that focused on migrants in Western countries (*Yeasmin & Regmi, 2013*; *Manderson & Mathews, 1981*). We excluded this second group of studies in order to avoid reliance

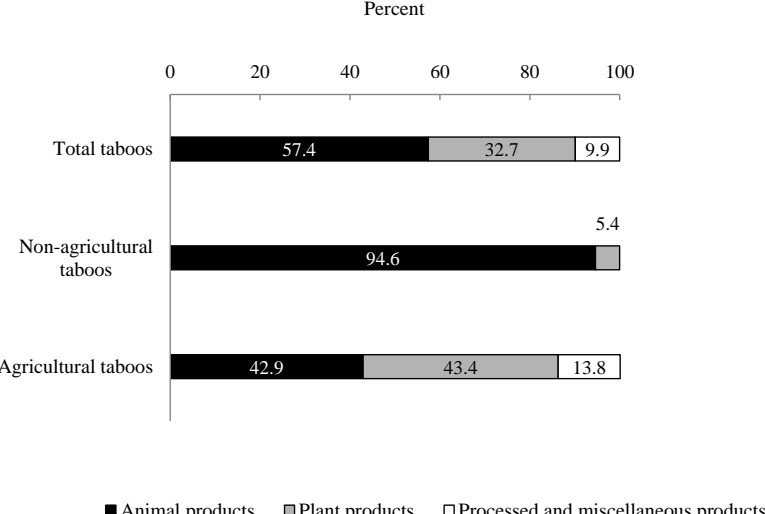

**Distribution of food types avoided during pregnancy**

■ Animal products  ▨ Plant products  ☐ Processed and miscellaneous products

**Figure 2** **Percentage distribution of food types avoided during pregnancy.** The first bar in the graph refers to the *Food type* distribution among total taboos. The remaining two bars show the *Food type* distribution within agricultural and non-agricultural taboos. The percentages of *Animal products* (black), *Plant products* (grey) and *Processed and miscellaneous products* (white) differed significantly between agricultural and non-agricultural taboos, ($X^2 = 58.433$, df $= 2$, $p < 0.001$, Cramer's $V = 0.471$; $n = 263$). *Plant and processed products* are more present among the taboos of agriculturalists, and *Animal products* are more present among those of non-agriculturalists.

on non-farming and non-foraging occupational activities, industrial staples, medical guidelines, and loss of adherence to original cultural customs on dietary behaviour during pregnancy. Food taboos mentioned in the eligible studies but for which reasons were not clearly stated (*Ekwochi et al., 2016*; *Pritham & Sammons, 1993*; *Henrich & Henrich, 2010*) were also excluded from final analysis. Such exclusion did not affect the significance of the distribution of *Food type* across *Subsistence patterns,* but helped to perform and present analysis on *Food types* and *Food focus* based on equal group size (see *Differences in food types between subsistence patterns* and *Differences in taboo focus between subsistence patterns*).

The majority of food taboos during pregnancy were agricultural ($n = 189$, 71.9% of the total), with the remaining being non-agricultural ($n = 74$, 28.1%). Across all taboos, 57.4% targeted *Animal products*, 32.7% *Plant products* and 9.9% *Processed and miscellaneous products* (Fig. 2). Regarding the given reason, 38.8% focused on *No or unspecified physiological complications*, 36.9% on *Varied physiological complications*, and 24.3% on a *Big babies and difficult delivery* (Fig. 3).

## Differences in food types between subsistence patterns

There was a significant association between the *Food types* and *Subsistence pattern* variables. *Animal products*, *plant products*, and *processed and miscellaneous products* were significantly differently distributed between agricultural and non-agricultural taboos ($X^2 = 58.433$, df $= 2$, $p < 0.001$, Cramer's $V = 0.471$). Non-agricultural groups taboos showed a much greater

**Distribution of focus of food taboos during pregnancy**

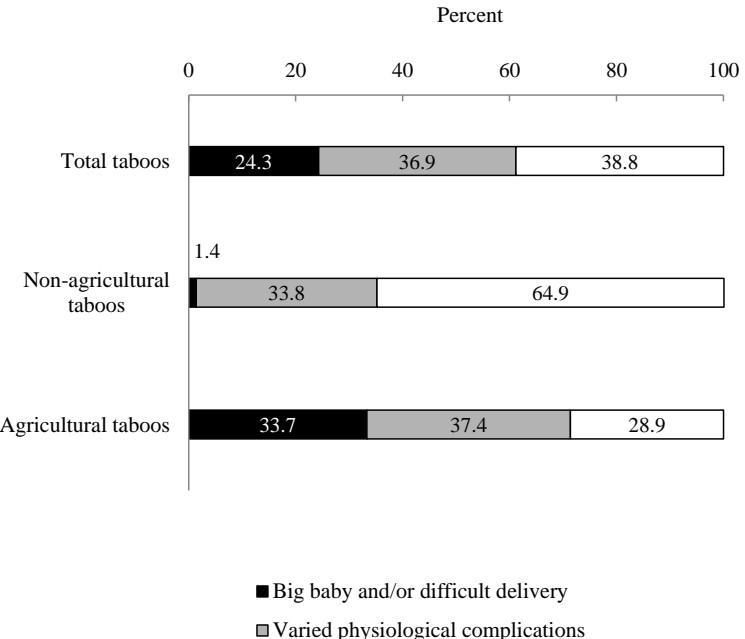

> **Figure 3  Percentage distribution of taboos focus.** The first bar in the graph refers to the distribution of the focus on the taboo, as a percentage of the total number of taboos. The remaining two bars show the distribution of the focus within agricultural and non-agricultural taboos. *No or unspecified physiological complications* (white) were significantly higher among non-agricultural taboo focus, while fear of *Big baby and/or difficult delivery* (black) was significantly higher among agricultural taboo focus, $X^2 = 40.682$, df = 2, $p < 0.001$, Cramer's $V = 0.393$ (N 263). No significant difference was found for the distribution of *Varied physiological complications* (grey) between subsistence patterns.
>
>

share of animal products (94.6%) compared to agricultural groups taboos (42.9%) (Fig. 2, Table 4). In contrast, agricultural taboos included more plant (43.4%), and processed and miscellaneous (13.8%) products than non-agricultural taboos (plant products: 5.4%; processed products: 0%). Animal products mentioned in taboos of non-agriculturalists were always non-domesticated species, whereas taboos from agriculturalists also included farmed animal products and dairy products (Table 2). The few plant products among taboos of non-agriculturalists referred to wild or occasionally cultivated tubers, or to a food's shape, while agricultural taboos encompassed products that were home-cultivated, or linked to intensive agriculture or industrial production chains (Table 2).

## Differences in taboo focus between subsistence patterns

There was a significant association between *Taboo focus* and *Subsistence pattern* ($X^2 = 40.682$, df = 2, $p < 0.001$, Cramer's $V = 0.393$, N 263). Non-agricultural taboos had a significantly higher proportion of reasons aimed at avoiding dangers less directly imputable to the physiological sphere (64.9%) compared to agriculturalists (28.6%)

**Table 5 Significant differences in focus of food taboos during pregnancy across subsistence modes.** The values (frequencies and percentages) in bold type are those that are significantly higher than the values in italics in the same row.

|  |  | Agricultural taboos | Non-agricultural taboos |
|---|---|---|---|
| Big baby and/or difficult delivery | N (%) | **63 (33.3%)** | *1 (1.4%)* |
| Varied physiological complications | N (%) | 72 (38.10%) | 25 (33.8%) |
| No or unspecified physiological complications | N (%) | *54 (28.6%)* | **48 (64.9%)** |
| Totals | N (%) | 189 (100%) | 74 (100%) |

**Table 6 Significant differences in taboo food types during pergnancy across focus of taboos.** The values (frequencies and percentages) in bold type are those that are significantly higher than the values in italics in the same row.

|  |  | Big baby and/or difficult delivery | Varied physiological complications | No or unspecified physiological complications |
|---|---|---|---|---|
| Animal products | N (%) | *22 (34.40%)* | *50 (51.5%)* | **79 (77.5%)** |
| Plant products | N (%) | **29 (45.30%)** | **37 (38.1%)** | *20 (19.6%)* |
| Processed/miscellanoeus products | N (%) | **13 (20.30%)** | 10 (10.3%) | *3 (2.9%)* |
| Totals | N (%) | 64 (100%) | 97 (100%) | 102 (100%) |

(Fig. 3, Table 5). The fear of gestating a big baby and/or a difficult delivery was in turn significantly more frequent among taboos from agricultural groups (33.3%) than among those of non-agriculturalists (1.4%). No significant difference in Varied physiological complications emerged, with this category representing the 38.1% of reasons given for agricultural taboos, and the 33.8% given for non-agricultural taboos.

## Differences in food types between categories of taboo focus

The distribution of *Food type* categories differed significantly between *Taboo focus* categories ($X^2 = 35.002$, df = 4, $p < 0.001$, Cramer's V = 0.258, N = 263) (Fig. 4, Table 6). The frequency of plant taboo was significantly higher in *Big babies and/or difficult delivery* (45.3%) and *Varied physiological complications* (38.1%) than into *No or unspecified physiological complications* (19.6%). By contrast, animal products were majorly present in the No or unspecified physiological complications category (77.5%) than in the *Big babies and/or difficult delivery* (34.4%) and *Varied physiological complications* (51.5%) categories. Processed and miscellaneous products were significantly majorly present in taboos justified by *Big babies and/or difficult delivery* (20.3%) than *No or unspecified physiological complications* (2.9%).

## Control analysis

*Food type* and *Taboo focus* differed significantly between *Subsistence patterns* in Africa ($p = 0.002$; $p < 0.001$) and Asia ($p < 0.001$; $p < 0.001$), while in Oceania differences were significant for *Food type* ($p = 0.0003$) but not *Taboo focus* ($p = 0.37$) (Figs. 5 and 6). The control analysis confirmed that, regardless of geographic distribution, agricultural taboos

**Distribution of food types avoided during pregnancy between categories of taboo focus**

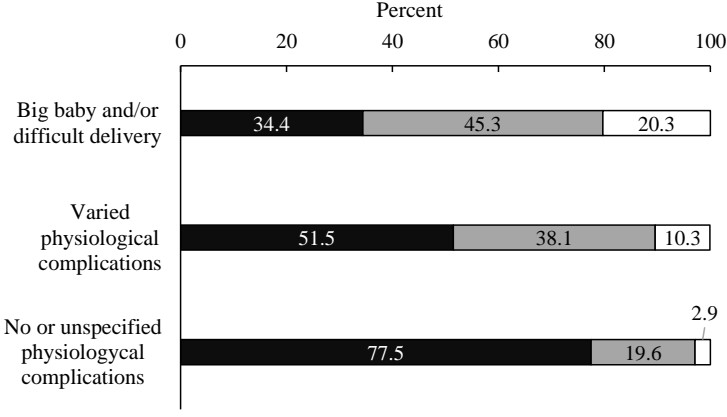

**Figure 4** **Percentage distribution of food types avoided during pregnancy between categories of *Taboo focus*.** *Animal products* (black) were significantly more present in the *No or unspecified physiological complications* category than in *Big baby and/or difficult delivery*. Percentages of *Plant products* and *Processed and miscellaneous products* were significantly higher in the *Big baby and/or difficult delivery* category than in *No or unspecified physiological complications*, $X^2 = 35.002$, df $= 4$, $p < 0.001$, Cramer's $V = 0.258 (N = 263)$.

were more likely to target plant and processed foods and to be motivated by the fear of *Big babies and/or difficult delivery*, while non-agricultural taboos were significantly associated with animal products and *No or unspecified physiological complications* (Figs. 5 and 6). The fact that a similar trend is shown in the distribution of *Taboo focus* in Oceania but not at a significant level may derive from the smaller number in the sample of taboos available for Oceania ($N = 36$) compared with Africa ($N = 62$) and Asia ($N = 142$). Therefore, the control analysis did not change the main findings.

## DISCUSSION

Our analysis has contributed three new findings. First, we found significant differences in the distribution of food types between agricultural and non-agricultural taboos. Agricultural taboos tend to target plant, processed and miscellaneous foods compared to non-agricultural taboos, which are primarily concerned with non-domesticated animal species. Second, the fear of a difficult delivery due to a big baby is strikingly more present among agricultural taboos than among non-agricultural taboos, which are majorly justified by no or unspecified physiological concerns. Third, the fear of increased birth weight is more commonly attributed to plant and processed foods compared with non-physiological concerns, which are more commonly associated with animal products.

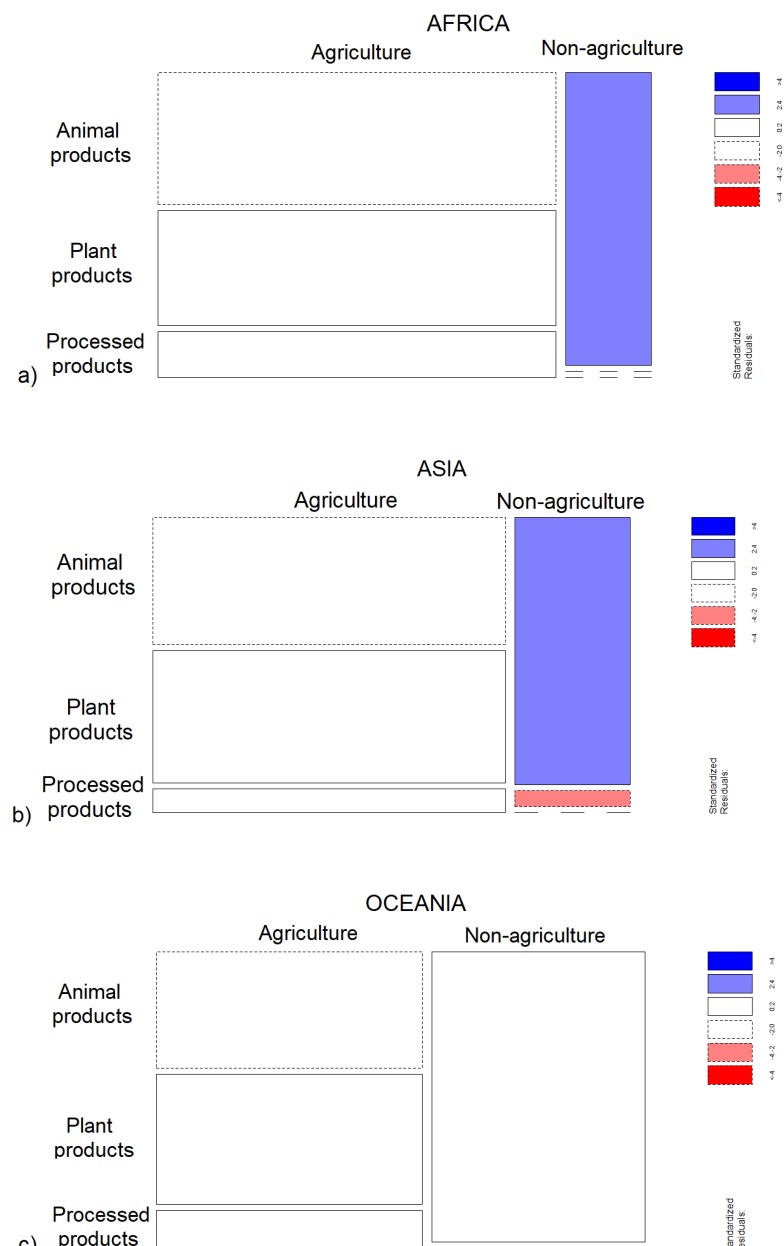

**Figure 5  Distribution of food types between subsistence patterns within geographical regions.** Mosaic plot that shows which cells contribute most to the significance of the test of independence (Fisher's Exact test) between subsistence patterns and taboo food types in (A) Africa, (B) Asia and (C) Oceania. The width of squares represents the numerosity of each subsistence pattern category, while the height the numerosity of each category of food type within each subsistence pattern. Pattern of blue (positive values of standardized Pearson residuals) show cells whose observed frequency is greater than would be found under independence. Pattern of red (negative values) show cells whose observed frequency is less than would be found under independence. The frequency of animal products is greater than would be found under independence among non-agriculturalists from (A) Africa and (B) Asia. White squares indicate positive

**Figure 5 (…continued)**
(solid line) and negative (dotted line) values of standardized Pearson residuals of cells whose observed frequency does not significantly differ from the distribution of data under independence. Plant and processed products are thus more frequently mentioned than animal products among agriculturalists in (A) Africa, (B) Asia and (C) Oceania, even if the distribution does not diverge sharply from the indipendent distribution. Similarly, non-agriculturalists in (C) Oceania mention animal products more frequently than plant and processed products but with lower standardized Pearson residuals than in (A) Africa and (B) Asia.

The *a posteriori* identification of a qualitative gradient in physiological specificness and the likelihood of risks addressed by taboos during pregnancy confirms our assumption that they flag actual and varied biological stresses, in addition to, or overlapping with, social and symbolic aspects. The *No or unspecified physiological complications* category of taboo focus exemplifies the tendency of many societies to ascribe health problems to troubled relations with the non-material world (*Voeks & Sercombe, 2000*), even though the avoidance of generic sickness and death, and physical-behavioural anomaly, is often the final aim (Table 3). For example, if pregnant Sago women of Papua New Guinea eat taboo foods they consider themselves generically at risk of being killed by the *aye ipari* spirits (*Townsend, 1971*), while the spirits of taboo animals among the Orang Asli are thought to produce *sawan*, an umbrella term used to indicate severe illness (*Bolton, 1972*). In Mornington Island, the consumption of unborn dugong foetus is banned to avoid children's weakness (*Pearn & Sweet, 1977*). Some of the undesired physical and behavioural characteristics listed in the *No or unspecified physiological complications* category appear to be around aesthetic effects arising through the perceived assimilation of a food's features by the new-born, rather than actual threats to health. For exampe, large fish may be avoided to avert an unsightly large mouth (*Pearn & Sweet, 1977*), armadillos avoided to prevent the baby acquiring its characteristic fear (*Holmberg, 1950*) and goats avoided to prevent wildness (*Oishi et al., 2000*). Ducks, chicken and rabbits are particularly avoided to prevent transmitting their general appearance to the child (*Pritham & Sammons, 1993*; *Gao et al., 2013*; *Oishi et al., 2000*). Without in depth research on what is considered beautiful, healthy or ideal for childbirth within each culture (*Simoons, 1994*; *Carles, 2014*; *Marcel, Justine & Florentine, 2015*), we cannot speculate on a generalized underlying health value of this category of taboos, nor on their rigid confinement within local socio-aesthetical standards around childbirth.

Leaving this debate open to future research, we can at least affirm that the threats listed in the *No or unspecified physiological complications* category have minor physiological likeliness than other undesired conditions mentioned in literature. Indeed, the *Varied physiological complications* category shows a clearer cross-cultural interest in protecting mothers, the childbirth process, and infants from dangers that are explicit but which are also plausible physiological risks (Table 3). Some of these complications, like nausea, haemorrhage, miscarriage, and malformations, fall under the domain of pregnancy-specific discomforts. Other complications, like skin, gastrointestinal and respiratory problems, can affect any life stage. Even if fear of miscarriage, malformations and skin problems are mentioned more frequently, we were unable to detect any numerically consistent semantic
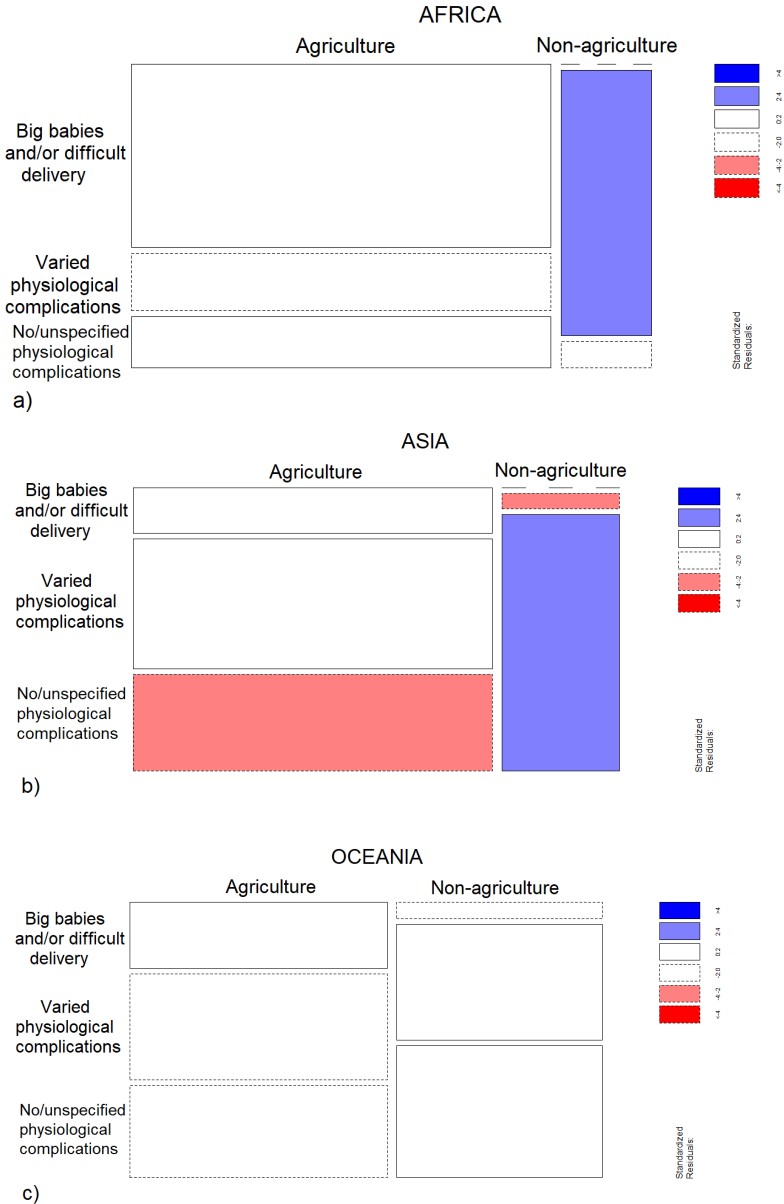

**Figure 6** **Distribution of focus of taboos between subsistence patterns within geographical regions.**
Mosaic plot that shows which cells contribute most to the significance of the test of independence (Fisher's
Exact test) between subsistence patterns and focus of taboos in (A) Africa, (B) Asia and (C) Oceania. The
width of squares represents the numerosity of each subsistence pattern category, while the height the nu-
merosity of each category of taboo focus within each subsistence pattern. Pattern of blue (positive values
of standardized Pearson residuals) show cells whose observed frequency is greater than would be found
under independence. Pattern of red (negative values) show cells whose observed frequency is less than
would be found under independence. (A) Africa: the frequency of varied physiological complications is
greater than would be found under independence among non-agriculturalists. White solid line squares in-
dicate that fear of big babies, but also the less frequently mentioned no or unspecified physiological com-
plications, show positive though not extreme values of standardized Pearson residuals among agricultural-
ists. (continued on next page...)

**Figure 6 (…continued)**
(B) Asia: No or unspecified physiological complications are mentioned more frequently than under independence among non-agriculturalists. Conversely, this taboo focus is mentioned less frequently than under independence among agriculturalists (red square), where frequency of fear of big babies and varied physiological complications show positive though not extreme values of standardized Pearson residuals (white and continuous line squares). (C) Oceania: agricultural taboos show high frequency of fear of big babies, while non-agricultural taboos show high frequency of varied and no or unspecified physiological complications, even if this distribution does not divergence significantly from that expected under independence between variables.

sub-cluster within *Varied physiological complications* that could form a separated focus category. A larger ethnographic database on food taboos during pregnancy and their reasons would provide the opportunity to identify more categories of reasons associated with hazards that are both specific and numerically consistent. While future research may address this gap, we can at least interpret the *Varied physiological complications* category identified in this study as the presence of a large share of cross-cultural taboos that are intended to protect maternal and foetal biology but in very diverse ways.

Moreover, we cannot propose a general link between all *Varied physiological complications* (Table 3) and the foods identified as causal agents (Table 2). Rather, the *Varied physiological complications* category exemplifies that at least three non-mutually exclusive scenarios seem to exist across cultures that embody the varied health value of taboos during pregnancy. In the first scenario, taboos are recognized agents of an observed discomfort, and such recognition has validity in scientific terms, as with the case of Fijian taboos against symptoms of food poisoning which target actual toxic species (*Henrich & Henrich, 2010*). In the second scenario, the scientific validity of the recognised relationship is plausible but not easy to test, as for example the teratogenic effects of some plant products that popular Indian beliefs link with abortion or malformations (*Ferro-Luzzi, 1973*; *Placek, Madhivanan & Hagen, 2017*). In the third scenario, taboos still address plausible biological discomforts, but their nutritional cause, which may be absent in scientific terms, may be locally identified through aesthetical analogies or symbolism. An example of latter scenario is the attribution of cleft-lip to the consumption of rabbit (*Mustafina et al., 2019*).

The fact that the above-mentioned taboos may not be effective at achieving their intent is explained in the wider discourse around the different drivers of transmission of general taboos (*Navarrete & Fessler, 2003*). However, this does not preclude their final aim being to avert biological complications, which helps to detect cross-cultural perceived threats to maternal-infant health. In contrast to the apparent discordance of reasons labelled as *Varied physiological complications*, the high frequency of taboos motivated by the fear of *Big babies and/or difficult delivery* allowed us to create a separated focus category. This shows that a consistent share of cross-cultural taboos addresses a risk that is both specific to childbirth, and also clinically plausible in its recognised association with maternal diet.

A causal relation between food perceived as fattening, increased infant size and difficult delivery has been reported by women from agricultural communities of Nigeria (*Ekwochi et al., 2016*), Ghana (*Dove, 2010*; *Otoo, Habib & Ankomah, 2015*), Ethiopia (*Demissie & Kogi-Makau, 1998*), Ivory Coast (*Donné Kouadio, 2017*), Tanzania (*Marchant et al., 2002*),

Tajikistan (*McNamara & Wood, 2019*), Bangladesh (*Shannon et al., 2008*), India (*Ferro-Luzzi, 1973*), Thailand (*Liamputtong et al., 2005*), Indonesia (*Hartini, 2004*), Papua New Guinea (*Kuzma et al., 2013*), Laos (*Holmes et al., 2007*; *Daviau, 2003*), and Korea (Sich, 1981, as cited in *Pritham & Sammons, 1993*; *Oishi et al., 2000*). A difficult delivery caused by large infants belongs to the clinical spectrum of obstructed labour, that occurs when the passage of the foetus is mechanically obstructed due to incompatibility of the physical size of the mother and/or the foetus (*Konje & Ladipo, 2000*). Interestingly, obstructed labour is a common cause of mortality in West Africa, India, Bangladesh and Tajikistan (*Ould El Joud et al., 2002*; *Sikka et al., 2011*; *Coyaji, 1991*; *Alauddin, 1986*; *Wiegers, Boerma & de Haan, 2010*), and generally in countries with limited access to health care (*Rush, 2000*).

Moreover, the fear of increased birth weight is attributed to foods such as bananas, ripe fruits, sugarcane, starches, baked and sugary items, but also to a minor share of meat, eggs, milk and fish (Table 2, Fig. 4). These foods either have medium to high glycaemic index according to international tables (*Atkinson, Foster-Powell & Brand-Miller, 2008*), or are protein sources. Human variance in birth weight largely depends on maternal dietary carbohydrate, and moderate-to-high glycaemic foods increase foetal weight (*Clapp III, 2002*; *Moses et al., 2006*; *Moses et al., 2014*), while protein contribute to fetal weight gain during the first half of gestation (*Rao, Shashidhar & Ashok, 2013*).

Causes of obstructed labour are not only nutritionally induced foetal growth acceleration and high birth weight and head circumference, but also maternal traits including diabetes mellitus, young age and primiparity, short height and narrow birth canal (*Konje & Ladipo, 2000*; *Rush, 2000*; *Tsur et al., 2012*; *Wells, 2017*, *Bochner et al., 1987*; *Casey et al., 1997*). Interestingly, these factors also appear to overlap with the areas where taboos against difficult delivery are present. The incidence of maternal gestational diabetes, linked to dystocia and foetal macrosomia, is rising in South Asia, South East Asia, and Africa (*Jones, 2001*; *Yuen & Wong, 2015*; *Muche, Olayemi & Gete, 2019*). Moreover, short adult stature, derived from child stunting, exacerbates the impact of maternal height on childbirth complications in India and other countries affected by malnutrition (*Wells, Wibaek & Poullas, 2018*; *Wells et al., 2021*). Not surprisingly, the advice to 'eat down' during pregnancy that is popular in Laos, Bangladesh, Pakistan and India, is addressed especially to small women (*Karim et al., 2002*; *Hutter, 1996*; *Rush, 2000*; *Holmes et al., 2007*; *Asim et al., 2021*; *Harding et al., 2017*; *Yeasmin & Regmi, 2013*; *Shannon et al., 2008*; *Nag, 1994*; *Nichter & Nichter, 1983*).

Traditionally, evolutionary life history theory has largely focused on overall energy availability as the key component of nutrition subject to selection (*Hill, 1993*), and broadly assumes that greater energy intake benefits maternal fitness through increasing both fertility and the quality of each offspring. On average, larger mothers deliver larger babies, mediated in part by larger dimensions of the birth canal (*Wells, Figueiroa & Alves, 2017*). However, childbirth offers a unique content in which greater foetal growth may threaten the fitness of both mother and offspring (*Wells, DeSilva & Stock , 2012*), as demonstrated in recent studies of populations where fetal size is increasing over time (*Wells, Wibaek & Poullas, 2018*), and this links with increasing awareness that the balance of substrates in foods have metabolic implications beyond their total caloric supply (*Moses et al., 2006*; *Moses et al.,*

*2014*). Likewise, the extensive literature relating pregnancy taboos to specific foods, rather than to overall diet, suggests that this understanding is socially embedded in numerous settings.

The results of the present study thus support *Dove (2010)* and Rush's (*2000*) observations that the preference for the delivery of smaller children embodied by food taboos reveals a serious concern that large babies put mother and infants at risk in Africa and Asia. The finding of significant differences between agricultural and non-agricultural taboos, supporting our initial hypothesis that subsistence modes can explain part of cross-cultural diversity in food avoidance during pregnancy, further confirms that the fear of obstructed labour represents a new window on several clinical and evolutionary issues.

On one hand, our findings suggest that a broadened pool of agricultural staples in terms of cultivated and processed products could have expanded the pool of agricultural taboos, while those of non-agriculturalists seem to remain concentrated on diverse non-domesticated animal species. On the other hand, the significant asymmetry in *No or unspecified physiological complications* among non-agricultural taboos and the very specific fear of *Big babies and/or difficult delivery* in agricultural taboos couples with the absence of significant differences in *Varied physiological complications*. This seems to indicate that the intent to protect from varied physical dangers is equally present in agricultural and non-agricultural taboos, but the specific concern on obstructed labour due to large infants is, if not typical of, at least more likely to spread in agricultural contexts. Indeed, a concern over increased birth weight is mentioned only once in the literature on hunter-gatherers in relation to the consumption of too much meat among the Ngatatara of Central Australia (*Roheim, 1933*), and there is evidence that among contemporary forager populations, childbirth is not characterized by difficulties during delivery or exceptional birthweights (*Howell, 2010*, p. 23; *Roy, 2003*).

The contrast, in our results, of fear of increased birth weight and incidence of obstructed labour between agricultural and non-agricultural contexts resonates with the evolutionary argument that agricultural high glycaemic foods may have exacerbated childbirth difficulty both by enlarging baby size, and by decreasing stature during the Neolithic (*Wells, DeSilva & Stock , 2012*). This scenario might occur repeatedly in different populations at different times, whenever environmental factors that promote birth weight, including nutrition-transition towards industrial foods, change at faster rates than those that regulate stature and/or pelvic maturity (*Wells, DeSilva & Stock , 2012*; *Dunsworth et al., 2012*). Moreover, the differential ability to clear blood sugar across modern populations, on which high glycaemic foods may have acted as selective agents, may explain higher contemporary rates of diabetes and obstructed labour in non-western countries, where exposure to high carbohydrate diets only occurred relatively recently (*Brown, Ruvolo & Sabeti, 2013*; *Fumagalli et al., 2019*).

As a final consideration, *Voeks & Sercombe (2000)* observed that Penan hunter-gatherers of Brunei, and perhaps other tropical forest foragers, have a medical system that is limited in size, scope and detail compared to neighbouring rice cultivators, probably because their foraging lifestyle generates limited exposure to infectious and nutritional diseases. Similarly, non-agricultural dietary changes during pregnancy seem not only to have reduced

physiological focus, but are also less widespread or dramatic than agricultural changes. Food taboos during pregnancy are not present among the African !Kung and Hadza, or some Asli groups and the RuMuda in Malaysia (*Howell, 2010*, p. 23; *Fitzpatrick, 2018*; *Bolton, 1972*; *Wilson, 1973*). They do not differ from those applied to other critical life-stages among the Sago of Papua New Guinea (*Townsend, 1971*), they seem to be less important than those during puberty among the Subarctic Athapaskan (*Asim et al., 2021*, as cited in *Spielmann, 1989*), and they are personal avoidances among the Mbuti (*Spielmann, 1989*). The taboo on deformed plants among the Sirono of Bolivia does not substantially impact the diet of pregnant women (*Holmberg, 1950*), whilst the animals prohibited among the Ntomba of Congo are rarely-consumed species (*Pagezy, 2006*). Therefore, we suggest that while non-agricultural food taboos during pregnancy do not seem to reveal any physiological stress on childbirth attributable to foragers' diet or lifestyle, the major share of taboos aimed at avoiding increased birth weight and difficult delivery among agriculturalists may result from the higher foetal growth potential of the agricultural diet that is responsible for a higher risk of obstructed labour.

To conclude, the present study showed that taboos during pregnancy are permeated with the symbolic and ecological value that connotates general taboos (*Golden & Comaroff, 2015*), but they also appear to specifically bring to the surface key issues in global maternal health, and the evolution of human childbirth in relation to that of diet. Our results contradict Fessler's finding that vegetables have minimal salience as a target of pregnancy taboos (*Fessler, 2002*), and they highlight that the avoidance of plant products has received little attention. Conversely, our study supports Fessler's assumption that meat is a central target of antenatal taboos because of an underlying ambivalence toward meat that evolved in humans, and not as direct consequence of maternal endogenous changes linked to food aversions (*Fessler, 2002*). Not controlling for subsistence patterns, the share of taboos that overlap with foods known to solicit aversion (*i.e.*, animals, Fig. 2) is not sufficiently common to infer that food aversions are the main driving force of food taboos. The general ambivalence toward animal foods, which may explain why these are the main target of general taboos for various specific reasons (*Navarrete & Fessler, 2003*; *Simoons, 1994*), may also explain the semantic diversity in *Varied physiological complications* and *No or unspecified physiological complications*, which are strongly associated with animal products (Fig. 4). Indeed, the anti-abortive and anti-food poisoning intent of antenatal dietary restrictions (*Henrich & Henrich, 2010*; *Placek, Madhivanan & Hagen, 2017*) are not as frequently, or as explicitly, mentioned as the specific intent to avoid a difficult delivery through the avoidance of carbohydrates. Therefore, the "fear of big babies" gains a new role in the evolutionary perspective on dietary restrictions during pregnancy, suggesting that obstructed labour has been perceived as a major life threat than a reduced diet during pregnancy.

The study had some strengths and weaknesses. Among the limitations are the lack of widespread data on actual maternal food intake and metabolic phenotype for each of the communities that showed taboos during pregnancy, and the methodological diversity of older accounts on food taboos among foragers. The generation of novel ethnographic material on restricted or changed dietary behaviour during pregnancy across cultures would

allow more nuanced interpretation of the data that our study aggregated in the *Varied physiological complications* and *No or unspecified physiological complications* categories. Future research is therefore needed to investigate more robust clusters of biological and non-biological rationales for taboos. Moreover, extending the investigation to nutritional surveys and patterns of foetal growth would allow assessment of whether the increase of neonatal size due to carbohydrate consumption has driven food taboos in order to reduce the risk of obstructed labour, especially in settings with high availability of cereals, starches, and industrial processed food products. In particular, societies undergoing rapid exposure to market economics and associated nutrition transition would represent a fertile research opportunity. Table 2 shows that industrial products as *sugary foods, wheat bread, noodles, ice cream, sugar* arrive in certain rural areas or smaller urban centres and these contribute to a perception of big babies (*Dove, 2010*; *Demissie & Kogi-Makau, 1998*; *Gao et al., 2013*; *Ferro-Luzzi, 1973*; *McNamara & Wood, 2019*), so that the marketing of industrial products may have exacerbated any tendency to associate foods with difficult delivery that was already present in non-industrialized agricultural contexts. This role of industrial foods can only be inferred and not demonstrated by our data, with support from the studies on economic and dietary shifts, and obstetric complications of *Ulijaszek, Mann & Elton (2012)*, *Cordain et al. (2005)*, *Wells, DeSilva & Stock (2012)*. However, the novel outlining of the fear of difficult delivery as a relevant cross-cultural driver of antenatal food restrictions matches growing evidence of the presence of factors of obstructed labour in Africa and Asia. Among the strengths are that the descriptive data available on non-agricultural taboos were enough to reconstruct a varied array of tabooed species in Africa, America, South East Asia and Australia, and to highlight an explicit, noteworthy absence of antenatal dietary restrictions among some strict hunter-gatherers societies.

## CONCLUSIONS

We analysed currently available literature on taboos during pregnancy to identify cross cultural taboo food types and focus. We tested the hypothesis that food types and focus differed between agricultural and non-agricultural taboos, and that they were associated one to each other. Non-agricultural taboos were more likely to target non-domesticated animals and to be justified by concerns not directly linked to the physiological sphere than agricultural taboos. By contrast, agricultural taboos addressed more cultivated and processed products and showed higher association to concerns over increased birth weight and difficult delivery. Overall, the fear of increased birth weight and difficult delivery was more frequently attributed to plant and processed products than to *No or unspecified physiological complications*. The widespread concern over big babies and carbohydrate rich foods overlaps with clinical evidence that obstructed labour is a major threat to maternal life in non-western countries. Moreover, the asymmetrical distribution of such concern between subsistence modes resonates with the possible impact of agricultural diet on delivery complications at an evolutionary level. Taboos to limit infant birth weight might be therefore a function of the foetal growth potential of the staple diet, namely of the agricultural subsistence mode. Limitations of this study are diversity in

methodological accuracy of current literature on food taboos during pregnancy, and the lack of comprehensive data on actual food intake and metabolic phenotype for each population under study. Future research is needed to (i) provide an accurate description of antenatal dietary behaviour both among individual populations and at a comparative level; (ii) investigate relations between attitude towards carbohydrate consumption during pregnancy, population-level incidence of obstructed labour, trends in stature and metabolic response to carbohydrate intake, and the glycaemic index of local staples.

### Funding

Funding for this study was provided by the School of Anthropology and Conservation at the University of Kent. The funders had no role in study design, data collection and analysis, decision to publish, or preparation of the manuscript.

### Competing Interests

The authors declare there are no competing interests.

### Author Contributions

- Ornella Maggiulli conceived and designed the experiments, performed the experiments, analyzed the data, prepared figures and/or tables, authored or reviewed drafts of the article, and approved the final draft.
- Fabrizio Rufo performed the experiments, authored or reviewed drafts of the article, and approved the final draft.
- Sarah E. Johns conceived and designed the experiments, authored or reviewed drafts of the article, and approved the final draft.
- Jonathan C.K. Wells performed the experiments, authored or reviewed drafts of the article, and approved the final draft.

### Data Availability

The raw data is available in the Supplementary File.

### Supplemental Information

Supplemental information for this article can be found online at http://dx.doi.org/10.7717/peerj.13633#supplemental-information.

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
