# Peer review of "Food taboos during pregnancy: meta-analysis on cross cultural differences suggests specific, diet-related pressures on childbirth among agriculturalists"

_PeerJ, doi:10.7717/peerj.13633_

## Round 0.1 · original submission · Major Revisions

Please address the concerns of both reviewers and amend the manuscript accordingly.

Reviewer 1 ·

Basic reporting

This paper makes an important contribution to the literature by focusing specifically on food taboos in agricultural societies in a cross-cultural perspective. The manuscript needs significant revisions, particularly in the analyses and background literature (see comments below).

1. The English used in the manuscript should be improved to enhance readability of the text. There are several instances of minor grammatical errors. Below are a few examples, but this is not an exhaustive list.
• Ln 29: should be data “were analyzed” not “was analyzed”
• Ln 46: “mean” should be “means”
• Ln 56: comma should be inside the quotations
• Ln 147: “human” should be “human’s”
• Ln 168: “we used reliance on” should be “we relied on”

2. In the opening paragraphs about evolutionary theory and pregnancy diet, the authors should discuss and cite the classic works of Margie Profet, Sherman, Flaxman, and Fessler, since they were the “trailblazers” (so to speak) on this topic, and formulated a model referred to as the “maternal fetal protection” hypothesis. See references below.

Flaxman, S. M., & Sherman, P. W. (2000). Morning sickness: a mechanism for protecting mother and embryo. The Quarterly review of biology, 75(2), 113-148.

Fessler, D. T. (2002). Reproductive immunosuppression and diet: an evolutionary perspective on pregnancy sickness and meat consumption. Current anthropology, 43(1), 19-61.

Profet, M. (1992). Pregnancy Sickness as Adaptation: A Deterrent to Maternal Ingestion. The adapted mind: Evolutionary psychology and the generation of culture, 327.


3. The reference for “eating down” is dated (ln 60-62). Please provide a more up-to-date reference if possible, because some sources suggest that it is no longer as common as once though (especially in India). Below are two references that discuss eating down and might have some relevant references cited within. One of them is a recent paper on pregnancy fasting, which provides more nuance to the practice of eating down.

Chanchani, D. (2019). Maternal and child nutrition in rural Chhattisgarh: the role of health beliefs and practices. Anthropology & Medicine, 26(2), 142-158.

Placek, C. D., Jaykrishna, P., Srinivas, V., & Madhivanan, P. (2021). Pregnancy Fasting in Ramadan: Toward a Biocultural Framework. Ecology of Food and Nutrition, 60(6), 785-809.

4. The post hoc analysis can be presented later, after the other results are presented. There is no need to put that information on ln 119.
5. I don’t think the information about this being a dissertation project is necessary for publication purposes (123-128).
6. Taboo reasons: why did the authors only choose those categories? The first one (big baby/difficult delivery) is very specific while the other two categories are not. Did abortion come up? Or what about specific physical abnormalities?
8. After reading the entire paper, I think the authors should cite the references I mentioned above about eating down and fasting, as they are highly relevant to the findings of the study.

Experimental design

N/A

Validity of the findings

I am a little confused about how the authors analyzed the data. Was it on the studies, or the taboos themselves (which equaled 263). If it is the latter, then I would like to see more sophisticated statistical analyses that control for geographical region.

Reviewer 2 ·

Basic reporting

This is well written. The literature review is generally sufficient, though I would like to see more included on food aversions. The article is well structured and presented.

Experimental design

This is adequate, though again, aversions could be interesting (if not necessary. The results are well-presented, especially considering that they are not statistically complex (a breath of fresh air).

Validity of the findings

The findings appear valid, though they could be presented perhaps more holistically. This is a nice article.

Additional comments

Major comments
1. I agree that this is an under-studied topic and yet certain references are missing, e.g., to Profet and Flaxman and Sherman, that speak to the importance of individual food aversions, as opposed to cultural taboos. The likely overlap in functions of aversions and taboos could be discussed with greater clarity for novice readers. The focus on subsistence is novel and interesting; it seems almost silly in hindsight that it has not been pursued previously.
a. Could search terms include items related to food aversion, in addition to taboos?
2. It might be interesting to see more of the qualitative results and/or lower N associations. There are good reasons to avoid meat and other animal products during pregnancy, none of which are described here, presumably because they are lumped under ambiguous physiological outcomes. The reasons for these taboos are thus obscured by the binning/aggregation of data. I’d like to see some of these, even if they are not central to the paper.
3. Given that the focus of this paper is on cultural taboos, it might be nice to see some consideration of non-health-related reasons to engage in taboos. Are there any relationships between taboos and family composition? Patri/matriliny? Food availability/insecurity? Signaling?
4. Relatedly, while the contribution of (over)nutritional risk is important, the article should probably also consider risks associated with epidemiological environments and energetic balance. This would certainly present a more thorough, life-history-oriented framing.

Minor comments
5. Line 46: Are all taboos concerned with dangers? Are they sometimes about control (e.g., menstrual huts/associated taboos)?
6. 123-8: I found this somewhat jarring – do we need to know about the history of supervision on this project?
7. 140 ff: Do any of the papers refer to non-industrialized versus industrialized agricultural contexts? Would market-based subsistence add a useful test of ideas presented about carbohydrates and baby size?
8. 201 ff: Should “no” and “unspecified” physiological complications be separated?
9. 257: The proportion of animal versus plant taboos is roughly identical in agricultural populations. Why should this be, if animal products are likely safer and/or concerns are related to the size of the baby?
10. 360-3: Surely, maternal energetic output matters here, too? Is it really only carbohydrate intake that determines the size of the baby?
11. 364 ff imply to me a sort of ideological catch-up to mismatch between nutritional surplus and fetal nourishment. In other words, I would more generally expect mothers to be encouraged to eat more, not less, during pregnancy, given the extra energetic burden.

---

## Round 0.2 · Minor Revisions

Please address the remaining concerns of the reviewer and revise the manuscript accordingly.

Reviewer 2 ·

Basic reporting

Improved, but could alter the text in the manuscript to situate more solidly within existing literature, rather than just adding citations.

Experimental design

Adequate, but subject to limitations, which are generally under-described, both in the first iteration and in this revision.

Validity of the findings

See above. Limitations require more thorough discussion.

Additional comments

Notes – Peer J Revision 68928

Abstract “useful for detecting biological pressures” is less cautious than it should be given the limitations acknowledged in the response to review. The data are suggestive of perceptions of problems, but not necessarily biologically meaningful ones, particularly given the tenuous links between nutrition and actual outcomes.

l.45-6: Pregnancy is more than just a period that ensures successful delivery; it sets the stage for later development, etc.

l.60-1: It would still be useful to characterize the prevalence of “eating down” versus “eating up” for audiences that aren’t familiar with these specific arguments. There are a lot of cultures where women have specific avoidances but are not encouraged to eat less overall.

l.67-9: I agree – these are not mutually exclusive interpretations of patterns, but it is probably more balanced to acknowledge the possibility that the cultural taboos do not necessarily have an obvious biological function – at least not a direct one.

l.80-1 – Again, agreed – but this has implications beyond just successful delivery.

l.87 ff – The new text is helpful here, but I think it is better to consider the possibility that aversions/fasting may serve different purposes in different times and places and the same purposes in any given context.

l.175-8 – I understand aggregating categories due to small N (as per the response letter), but it would be good to address associated limitations both early and late in the paper. “non-agriculture” is a highly heterogeneous bin consisting of potentially very different hazards for pregnant women and infants. This should be acknowledged and discussed.

Methods, general comment: It’s always a little disappointing to me when authors do not investigate alternative analyses or methods, particularly when reviewers converge on similar issues. The comments regarding the narrowness of the search from both reviewers suggest to me that a wider base for meta-analysis is appropriate and would resolve certain issues associated with sample size as well as how this work is situated within relevant literatures. There is very limited engagement with reviewer comments in the work presented, despite responses in the review.

Results: The “control analysis” if I understand it correctly investigates associations across three different gross geographic areas. What motivates these bins? And why is this geographic aggregation the only thing controlled for? Why not just present controlled analyses from the get-go?

Discussion: This does not do enough to incorporate the authors’ responses to review. Changes are very minimal compared to justifications in the response letter.

Summary: I like this paper and think it makes a valuable contribution, but I still think it could be stronger. Even if the authors do not want to adjust methods and analysis, they should be more cautious in their interpretations of the data, guiding future researchers to areas of nuance that remain under-investigated here.

---

## Round 0.3 · accepted · Accept

All remaining critiques were adequately addressed and the manuscript was revised accordingly. Therefore, this version is acceptable now.